# The sFlt-1/PlGF Ratio in Pregnant Patients Affected by COVID-19

**DOI:** 10.3390/jcm12031059

**Published:** 2023-01-29

**Authors:** Katarzyna Kosinska-Kaczynska, Ewa Malicka, Iwona Szymusik, Norbert Dera, Michal Pruc, Stepan Feduniw, Zubaid Rafique, Lukasz Szarpak

**Affiliations:** 1Department of Obstetrics, Perinatology and Neonatology, Center of Postgraduate Medical Education, 01-813 Warsaw, Poland; 2Research Unit, Polish Society of Disaster Medicine, 05-816 Warsaw, Poland; 3Department of Gynecology, University Hospital Zürich, 8005 Zürich, Switzerland; 4Henry JN Taub Department of Emergency Medicine, Baylor College of Medicine, Houston, TX 77030, USA; 5Institute of Outcomes Research, Maria Sklodowska-Curie Medical Academy, 00-136 Warsaw, Poland

**Keywords:** COVID-19, SARS-CoV-2, sFlt-1/PlGF ratio, RAS, placental dysfunction, pregnancy

## Abstract

COVID-19 in pregnant women increases the risk of adverse pregnancy outcomes, including preeclampsia. This meta-analysis aimed to examine the effect of SARS-CoV-2 infection on sFlt-1/PIGF ratio during pregnancy. The study was designed as a systematic review and meta-analysis. PubMed, Web of Science, Embase and Cochrane Library were searched for relevant studies reporting the sFlt-1/PlGF ratio in pregnant women with COVID-19. Results were compared using meta-analysis by the Mantel–Haenszel method. A total of 7 studies were included in the analysis. sFlt-1/PlGF ratios between COVID-19 positive vs. negative women were 45.8 ± 50.3 vs. 37.4 ± 22.5, respectively (SMD = 1.76; 95% CI: 0.43 to 3.09; *p* = 0.01). sFlt-1/PlGF ratios between asymptomatic vs. symptomatic patients were 49.3 ± 35.7 vs. 37.1 ± 25.6 (SMD = 0.30; 95% CI: −0.35 to 0.95; *p* = 0.36). sFlt-1/PlGF ratio in non-severe group was 30.7 ± 56.5, compared to 64.7 ± 53.5 for severe patients (SMD = −1.88; 95% CI: −3.77 to 0.01; *p* = 0.05). sFlt-1/PlGF ratios in COVID-19 patients, with and without hypertensive disease of pregnancy, were 187.0 ± 121.8 vs. 21.6 ± 8.6, respectively (SMD = 2.46; 95% CI: 0.99 to 3.93; *p* = 0.001). Conclusions: Patients with COVID-19, as compared to patients without COVID-19, were characterized by higher sFlt-1/PlGF ratio. Moreover, severe COVID-19 and SARS-CoV-2 infection in hypertensive pregnant women was related to significantly higher sFlt-1/PlGF ratio.

## 1. Introduction

Since its outbreak, the Coronavirus disease 2019 (COVID-19) pandemic, which was brought on by the severe acute respiratory syndrome coronavirus 2 (SARS-CoV-2), has been the cause of millions of deaths and has risen to the top of the list of causes of maternal mortality globally [1,2,3,4]. Nevertheless, its influence on pregnancy and the developing fetus has not been fully elucidated. It has been demonstrated that COVID-19 in pregnant women increases the risk of adverse pregnancy outcomes, including preeclampsia (PE), fetal growth restriction (FGR) and preterm birth [5,6,7]. The common cause of these pathologies has been proven to be placental dysfunction due to perfusion changes, infarcts, vasoconstrictions, decidual arteriopathy, intervillous thrombi, fibrin deposits or the presence of inflammatory infiltrates [8,9].

Perinatal outcome is related to the function of the placenta. SARS-CoV-2 influences the release of cytokines and growth factors. Vascular endothelial growth factor (VEGF) levels were found to be significantly increased in patients with COVID-19 compared to controls [10]. The biological activity of VEGF is regulated by a soluble portion of the fms-like tyrosine kinase (sFlt-1) receptor, produced mainly by the placenta and upregulated by placental dysfunction [11]. Placental growth factor (PlGF) is another protein that influences VEGF angiogenesis; its concentrations are also related to inappropriate placental vascularization [12]. The NICE guidelines suggest the usefulness of the sFlt-1/PlGF ratio as a PE predictor [13]. VEGF and PlGF interactions with receptors on the surface of vascular endothelial cells are blocked by sFlt-1. Consequently, VEGF and PlGF blood concentrations are decreased [12]. Failed placental vascularization leads to placental insufficiency, as the pathologic values of these markers are also seen in pregnancies with small for gestational age (SGA) and FGR fetuses [14].

Angiogenesis and placental invasion are also regulated by the Ren-in-Angiotensin system (RAS), which is responsible for renal and cardiovascular system function, in addition to being present in the placental tissue [15]. The RAS system plays an important role in trophoblast migration and is responsible for appropriate placental blood flow and fetal oxygen supply. Expression of the crucial Angiotensin-Converting Enzyme (ACE) decreases in the third trimester of physiological pregnancy. Analogous decreases have been observed earlier in pregnancies complicated by PE or FGR. Endothelial injury and hypoxia lead to the activation of the RAS system by means of Angiotensin II (ANG II) and its binding to the receptor. As a result, sFlt-1 production is increased. The RAS is also an important factor in the pathogenesis of COVID-19. The SARS-CoV-2 spike proteins bind to the ACE-2 in the cell membranes, permitting cellular invasion and the replication of the virus. The binding decreases the level of ACE-2. As ACE-2 plays an important role in protection against lung injury, the binding of the virus with ACE-2 influences the severity of COVID-19 [16,17,18].

The purpose of this meta-analysis was to examine previously reported findings on the effect of infection and the severity of COVID-19 on the sFlt-1/PIGF ratio during pregnancy.

## 2. Materials and Methods

This meta-analysis was conducted and reported in accordance with the Preferred Reporting Items for Systematic Reviews and Meta-Analyses (PRISMA) guidelines [19]. The study protocol was approved by all authors prior to the start of the study and the protocol was registered in the PROSPERO register (International Prospective Register of Systematic Reviews) under the number CRD42022381091.

### 2.1. Search Strategy

PubMed, Web of Science, Embase and Cochrane Library were searched from January 2020 to December 2022. The following combined search keywords were used in PubMed: “sFlt-1/PlGF” OR “Soluble fms-like tyrosine kinase-1” OR “Placental growth factor” AND “pregnancy” OR “pregnant” AND “COVID-19” OR “SARS-CoV-2” OR “severe acute respiratory syndrome coronavirus-2”. For other databases, the search terms were adapted correspondingly. For reports by the same author, only the latest or most intact were used to avoid the overlapping of queues. All references were imported into Endnote (ver. X9) and duplicates were removed before exporting them to the software-screening tool, Rayyan [20]. In addition, reference lists of relevant articles and systematic reviews were searched for further potential studies.

### 2.2. Inclusion/Exclusion Criteria

Studies meeting the following criteria were eligible for inclusion: (1) studies reporting the sFlt-1/PlGF ratio among pregnant women affected by COVID-19 vs. control group, or asymptomatic vs. symptomatic patients, or non-severe vs. severe group or COVID-19 patients with hypertensive disease of pregnancy (HDP) vs. non-HPD. Studies that met the following requirements were disqualified: (1) studies that failed to offer any of the results we had previously defined; (2) studies that lacked a comparable group; (3) studies not published in English; (4) publications such as editorials, conference papers, reviews, and letters to the editor.

### 2.3. Data Extraction

Data extraction was performed independently by two reviewers (M.P. and S.F.) using a prespecified data extraction form designed by L.S. The discussion with the third reviewer (L.S.) helped to settle any possible disputes between the other two. The following data were taken from the publications that qualified: study characteristics (first author, publication year, nation of origin, study design, research groups) and patient data (number of participants, age, and sFlt-1/PlGF ratios among research groups).

### 2.4. Bias Assessment

Two reviewers independently assessed the quality of the included studies. The third reviewer was consulted to settle any disputes between the reviewers (L.S.). The risk of bias within an individual cohort study was determined using the Newcastle Ottawa Scale (NOS) [21]. Three criteria were used by NOS to assess the quality of the study: selection, comparability and exposure. Each of the above three factors had maximum scores of 4, 2 and 3, respectively. High-quality studies were those with NOS ratings ≥ 7.

### 2.5. Data synthesis and Statistical Analysis

Data of selected studies were analyzed using Review Manager (ver. 5.4; Cochrane Collaboration, Oxford, UK). A *p*-value of <0.05 was considered statistically significant for all outcomes assessed. When a continuous outcome was reported in a study as median, range and interquartile range, means and standard deviations were estimated using the Hozo et al. formula [22]. The incidence of dichotomous data was performed using the odds ratio (OR) with 95% confidence interval (CI) and analyzed using the Mantel–Haenszel method. Continuous outcomes were described as standard mean difference (SMD) with 95% CI. Heterogeneity was assessed using the tau coefficient and measured using the I^2^ index; percentages of around I^2^ = 25%, I^2^ = 50% and I^2^ = 75% were considered low, medium and high heterogeneity, respectively [23]. The *p*-value ≤ 0.05 cut-point was used to declare statistical significance. Potential publication bias was assessed using funnel plots and, where possible, Egger’s regression test was performed. However, when a limited number of studies (<10) were included in the analysis, publication bias was not evaluated.

## 3. Results

### 3.1. Study Selection and Characteristics

Figure 1 describes a flow diagram summarizing the detailed steps of the study selection. In total, 519 articles were obtained through the database search. After removing duplicates, a total of 389 articles were screened for eligibility based on title and abstract. According to abstract and title evaluation, only 14 articles were assessed for full text screening. A total of seven studies were included in the meta-analysis [15,18,24,25,26,27,28]. Four studies reported the sFlt-1/PlGF ratio among COVID-19 positive patients vs. control group, two in asymptomatic vs. symptomatic groups, three in severe vs. non-severe COVID-19 and two among COVID-19 patients with hypertensive disease of pregnancy vs. non-HDP. Two studies were conducted in Mexico [15,25], two in Italy [24,27], and two in Spain [26,28]. One study was conducted in Poland [18]. The quality assessment of the studies (by the Newcastle Ottawa Scale) is presented in Table 1.

### 3.2. Meta-Analysis

Four studies reported sFlt-1/PlGF ratios between COVID-19 positive vs. negative patients. Pooled analysis showed that the sFlt-1/PlGF ratios between those groups were 45.8 ± 50.3 vs. 37.4 ± 22.5, respectively (SMD = 1.76; 95% CI: 0.43 to 3.09; *p* = 0.01; Figure 2). Pooled analysis of the sFlt-1/PlGF ratio between COVID-19 asymptomatic vs. symptomatic patients varied and amounted to 49.3 ± 35.7 vs. 37.1 ±25.6 (SMD = 0.30; 95% CI: −0.35 to 0.95; *p* = 0.36). The sFlt-1/PlGF ratio in non-severe group was 30.7 ± 56.5, compared to 64.7 ± 53.5 for severe COVID-19 patients (SMD = −1.88; 95% CI: −3.77 to 0.01; *p* = 0.05). Pooled analysis of the sFlt-1/PlGF ratio in COVID-19 patients with and without hypertensive disease of pregnancy varied and amounted to 187.0 ± 121.8 vs. 21.6 ± 8.6, respectively (SMD = 2.46; 95% CI: 0.99 to 3.93; *p* = 0.001).

## 4. Discussion

The presented meta-analysis investigated the influence of COVID-19 on the sFlt-1/PIGF ratio during pregnancy. The included studies found that sFlt-1/PlGF ratios in pregnant patients with COVID-19 were statistically significantly higher than in healthy pregnancies [18,26,27,28]. Moreover, there were no significant differences between ratios in symptomatic COVID-19 pregnant women and asymptomatic ones [24,28]. The sFlt-1/PlGF ratio was higher in pregnant patients with severe COVID-19 in comparison to those with non-severe illness [15,18,26]. In addition, hypertensive pregnant women with COVID-19 had higher sFlt-1/PlGF ratios than normotensive infected pregnant women [25,27].

A similar mechanism of placental injury during COVID-19 and preeclampsia is very probable. Mendoza et al. showed an increased risk of preeclampsia in patients suffering from COVID-19 [29]. Moreover, the correlation between COVID-19 severity and the occurrence of preeclampsia was presented in several studies [6,29]. Wei et al. confirmed an increased risk of severe PE (OR = 4.2, 95% CI: 1.6–11.2) in pregnancies with SARS-CoV-2 infection in their meta-analysis based on five studies. Furthermore, the risk of fetal growth restriction was also increased (OR = 1.9, 95% CI: 1.1–3.1) [30]. The probable common epithelial dysfunction in COVID-19 and PE could be caused by disturbances in the RAS system and, consequently, increased production of sFlt-1 and decreased PlGF concentrations. Espino-y-Sosa et al. found the sFlt-1/ANG-2 ratio to be a potential predictor of the severity of COVID-19 symptoms, but also a probable placental dysfunction marker [15]. Moreover, Hernandez-Pacheco et al. suggested a similar mechanism of increased sFlt-1/PIGF ratio in PE and COVID-19 [25].

SARS-CoV-2 infiltrates epithelial cells using surface spike proteins connected with ACE-2. Transmembrane Serine Protease 2 (TMPRSS2) of the virus destroys the S1-S2 junction, leading to the viral membrane and human cell fusion [31]. The subsequent formation of TLR2 heterodimers with TLR1 or TLR6 results in a complex of MyD88 and members of the IRAK kinases, which activates NF-B and MAPK signaling, causing the release of inflammatory cytokines and chemokines [32]. Similarly, in COVID-19, the expression of TLR is elevated in preeclamptic placentas [33,34]. It leads to cell destruction and immune system abnormalities. These changes, in turn, trigger the recruitment of functionally changed immune cells, altered cytokines, aberrant interferon-related responses and uncontrolled complement activation with concomitant neutrophil extracellular traps and systemic thrombosis. Complement-driven damage, a hyperinflammatory state defined by loss of T-cell subsets and elevated cytokine levels caused by IL-6, IL-1, TNF-alfa and humoral immunodeficiency with B-cell abnormalities, are direct causes of COVID-related cell damage [35]. Severe COVID-19 is related to excessively high levels of proinflammatory cytokines (IFN-gamma, TNF-alfa or IL-6) [36,37]. The hyperactivation of the RAS system is the result of the above-described hyperinflammation and hypercoagulability. During a typical pregnancy, RAS components are also found in the trophoblast, where they take part in trophoblast invasion, placental circulation and angiogenesis. A relative insensitivity to ANG II, which allows for low systemic vascular resistance, is typical for physiological pregnancy [38]. A few reports showed that inflammatory cascade reaction during severe SARS-CoV-2 infection injured the epithelium, as in PE. According to Giardini et al. this cytokine cascade could be present, not only in lung epithelium, but also in the placental cells [24].

The rates of miscarriage and intrauterine fetal demise are significantly increased in pregnancies with COVID-19 [7,39]. Schwartz et al. found pathological changes in the placentas of women with SARS-CoV-2 infection. Up to 78/% of placentas had increased fibrin deposition and necrosis or histiocytic intervillositis in trophoblasts [40]. As a result, inappropriate placental blood flow caused fetal hypoxia, leading to intrauterine fetal death or miscarriage. Direct fetal injury due to ongoing SARS-CoV-2 infection seemed less probable [7].

Information regarding the usefulness of the sFlt-1/PlGF ratio in the prediction of PE remains questionable, as SARS-CoV-2 infection also induces cytokine release in pregnant women. The diagnostic value of the sFlt-1/PlGF ratio could lead to false positives in pregnant patients with COVID-19. Regardless of COVID-19 symptoms, placental injury could already be present, as the level of viremia does not differ between symptomatic and asymptomatic patients [41]. Nevertheless, knowledge regarding these mechanisms could increase the survival rate of women and children. If the mechanisms of placental injury in PE and SARS-CoV-2 infection are similar, the prophylaxis could be efficient in both pathologies. Aspirintake, in pregnancy, was proven protective against impaired placentation [42]. It is possible that the intake of aspirin during SARS-CoV-2 infection in the first and early second trimester of pregnancy could lower the risk of PE and placental insufficiency [43]. However, further prospective studies are required to confirm this thesis.

The presented analysis had several limitations. The number of published articles on the correlation of the sFlt-1/PlGF ratio was small, and the study groups were not large. Therefore, the performed analysis could be influenced by the bias of the small population. Moreover, the articles were published between 2020 and 2022, which is a very short period of observation, and further follow-up research could bring more solid information. Another limitation could be the heterogenicity and mostly retrospective character of the included publications. Nevertheless, the observed trends clearly suggesedt that severe COVID-19 influenced the sFlt-1/PlGF ratio. Moreover, the presented study was the first meta-analysis comparing the sFlt-1/PlGF ratio and SARS-CoV-2 infection in pregnant women.

## 5. Conclusions

This meta-analysis showed the influence of COVID-19 on the sFlt-1/PIGF ratio during pregnancy. Patients with COVID-19, compared to patients without COVID-19, were characterized by higher sFlt-1/PlGF ratios. Severe COVID-19 and SARS-CoV-2 infection in hypertensive pregnant women was related to significantly higher sFlt-1/PlGF ratios. This wasx most probably due to similar mechanisms of placental damage and insufficiency in PE and COVID-19. Further prospective studies are required to confirm this thesis.

## Figures and Tables

**Figure 1 jcm-12-01059-f001:**
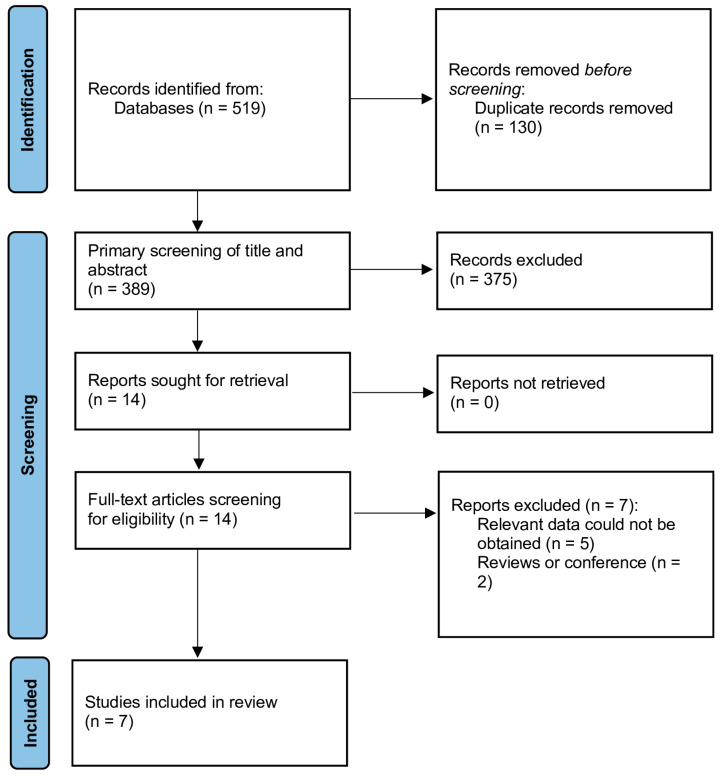
PRISMA flow diagram of the study selection process.

**Figure 2 jcm-12-01059-f002:**
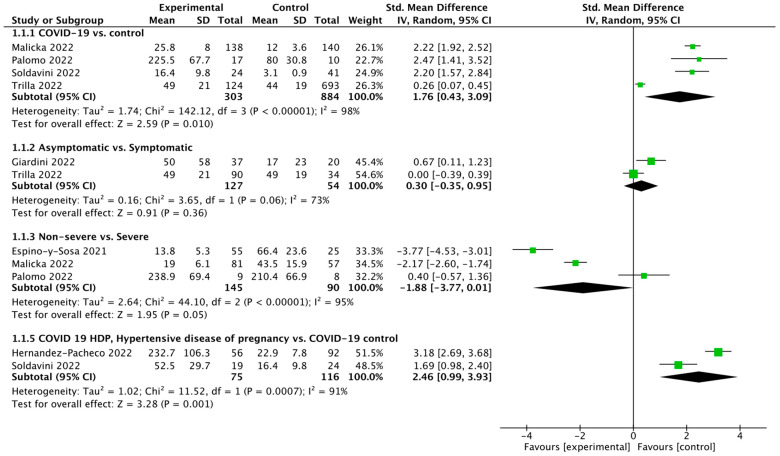
Forest plot of sFlt-1/PlGF ratio in different patients’ groups. The center of each square represents the standard mean differences (SMD) for individual trials, and the corresponding horizontal line stands for a 95% confidence interval (CI). The diamonds represent pooled results [15,18,24,25,26,27,28].

**Table 1 jcm-12-01059-t001:** Characteristics of included trials.

Study	Country	Study Design	Research Group	No. of Patients	Maternal Age	NOS Score
Espino-y-Sosa et al. 2021 [15]	Mexico	Prospective cohort study	Severe	25	30.8 ± 1.3	9
			Non-severe	55	29.1 ± 2.1	
Giardini et al. 2022 [24]	Italy	Retrospective trial	Asymptomatic	37	33 ± 5	8
			Symptomatic	20	33 ± 4	
Hernandez-Pacheco et al. 2022 [25]	Mexico	Multicenter retrospective cohort study	COVID-19 HDP	56	31.6 ± 2.4	7
			COVID-19 Non-HDP	92	30.2 ± 2.0	
Malicka et al. 2022 [18]	Poland	Case-Control Study	COVID-19 (+)	138	32.0 ± 1.3	8
			COVID-19 (−)	140	31.8 ± 1.5	
Palomo et al. 2022 [26]	Spain	Multicenter prospective population-based cohort study	Severe	8	33.6 ± 4.1	8
			Non-severe	9	35.1 ± 2.0	
			COVID-19 (+)	17	34.4 ± 3.2	
			COVID-19 (−)	10	36.0 ± 2.0	
Soldavini et al. 2022 [27]	Italy	Multicenter prospective population-based cohort study	COVID-19 HDP	19	34.8 ± 1.8	9
			COVID-19 Non-HDP	24	31.0 ± 1.5	
			COVID-19 (+)	24	16.4 ± 9.8	
			COVID-19 (−)	41	3.1 ± 0.9	
Trilla et al. 2022 [28]	Spain	Multicenter prospective population-based cohort study	COVID-19 (+)	124	33.1 ± 5.1	9
			COVID-19 (−)	693	33.9 ± 5.2	
			Asymptomatic	90	NS	
			Symptomatic	34	NS	

Legend: NOS: Newcastle Ottawa Scale; NS: Not specified.

## Data Availability

The data that support the findings of this study are available on request from the corresponding author (I.S.).

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
