# Peer review of "The sFlt-1/PlGF Ratio in Pregnant Patients Affected by COVID-19"

_jcm, 2023, doi:10.3390/jcm12031059_

Round 1

Reviewer 1 Report

This is a well-written paper with a well-designed method

I have a question about evaluating COVID-19 HDP, hypertensive disease of pregnancy vs. COVID-19 control. Indeed, the difference was due to hypertensive disease. Therefore, it seems a little pointless to me to present this result that has nothing to do with covid-19, deviating from what the study “The sFlt-1/PlGF ratio in pregnant patients affected by COVID-19: a systematic review and meta-analysis”, whose objective was “The purpose of this meta-analysis was to examine previously reported findings on the effect of infection and the severity of COVID-19 on the sFlt-1/PIGF ratio during pregnancy."

Author Response

I have a question about evaluating COVID-19 HDP, hypertensive disease of pregnancy vs. COVID-19 control. Indeed, the difference was due to hypertensive disease. Therefore, it seems a little pointless to me to present this result that has nothing to do with covid-19, deviating from what the study “The sFlt-1/PlGF ratio in pregnant patients affected by COVID-19: a systematic review and meta-analysis”, whose objective was “The purpose of this meta-analysis was to examine previously reported findings on the effect of infection and the severity of COVID-19 on the sFlt-1/PIGF ratio during pregnancy."

ANSWER: Thank you very much for your kind words about our work. The purpose of providing these data was to show that these biomarkers, despite the impact of COVID-19, are still prognostic and helpful in clinical practice. In this way, we also show the cut-off points that we should pay attention to when it comes to the levels of these biomarkers in COVID-19 patients, because the result alone in the case of the group without HDP could predict PE (if it was a patient without confirmed covid-19), which you probably know - it also shows reminding us that the increase in these biomarkers may not be caused only by COVID-19 - despite the significant difference in levels in both groups.

Reviewer 2 Report

In the manuscript "The sFlt-1/PlGF ratio in pregnant patients affected by COVID-19: a systematic review and meta-analysis" the authors systematically reviewed and analyzed available literature data on the effect of infection and the severity of COVID-19 on the sFlt-1/PIGF ratio during pregnancy. The authors very thoroughly analysed previously reported findings.

Major comment:  

In Table 1. the authors showed characteristics of included trials. Soldavini et al. analysed data from four groups of subjects: 19 COVID-19, HDP patients, 24 COVID-19, non HDP patients, 185 non COVID-19, HDP patients, 41 normotensive controls (non COVID-19, non HDP). These data are included in Table 1. but it appears that the number of non COVID-19 patients is obtained by summing HDP and non HDP uninfected subjects. Considering that sFlt-1/PlGF ratio is significantly higher in HDP in comparison to normotensive controls, please explain.

Minor comments:

Line 62 – please correct in-fluence

Line 94 – please correct re-porting

Line 309 – please add volume and manuscript number in reference list.      

Author Response

In the manuscript "The sFlt-1/PlGF ratio in pregnant patients affected by COVID-19: a systematic review and meta-analysis" the authors systematically reviewed and analyzed available literature data on the effect of infection and the severity of COVID-19 on the sFlt-1/PIGF ratio during pregnancy. The authors very thoroughly analysed previously reported findings.

Major comment:  In Table 1. the authors showed characteristics of included trials. Soldavini et al. analysed data from four groups of subjects: 19 COVID-19, HDP patients, 24 COVID-19, non HDP patients, 185 non COVID-19, HDP patients, 41 normotensive controls (non COVID-19, non HDP). These data are included in Table 1. but it appears that the number of non COVID-19 patients is obtained by summing HDP and non HDP uninfected subjects. Considering that sFlt-1/PlGF ratio is significantly higher in HDP in comparison to normotensive controls, please explain.

ANSWER: Thank you very much for this valuable comment, of course, an error crept in, which we corrected - these groups should not be added up because, as you well know, the HDP group would significantly inflate the results of patients in this group. We improved this and the results should be much better after excluding patients with HDP in this subanalysis due to higher scores for biomarkers known to us from, for example, PE diagnosis and prognosis.

Minor comments:

Line 62 – please correct in-fluence

Line 94 – please correct re-porting

Line 309 – please add volume and manuscript number in reference list.    

ANSWER: The editorial errors above have been corrected.

Round 2

Reviewer 2 Report

In the revised manuscript, the authors answered my concern. I have no further questions.